# Investigating the Thermo-Optic Properties of BCZT-Based Temperature Sensors

**DOI:** 10.3390/ma16145202

**Published:** 2023-07-24

**Authors:** Manlika Kamnoy, Kamonpan Pengpat, Tawee Tunkasiri, Orawan Khamman, Uraiwan Intatha, Sukum Eitssayeam

**Affiliations:** 1Department of Physics and Materials Science, Chiang Mai University, Chiang Mai 50200, Thailand; queen.n.mk@gmail.com (M.K.); kpengpat@gmail.com (K.P.); tawee.tun@cmu.ac.th (T.T.); orawankhamman@gmail.com (O.K.); 2School of Science, Mae Fah Luang University, Chiang Rai 57000, Thailand; uraiwan.int@mfu.ac.th

**Keywords:** photoluminescent, electroluminescent, thermo-optic properties, temperature sensors, spin coating method

## Abstract

Photoluminescent (PL) layers and electroluminescent (EL) systems have gained significant attention for their applications in constructing flat panels, screen monitors, and lighting systems. In this study, we present a groundbreaking approach to fabricating temperature sensors using barium-calcium zirconium titanate (BCZT) with thermo-optic properties, leading to the development of opto-thermal sensors for electric vehicle battery packs. We prepared zinc sulfide (ZnS) fluorescent films on BCZT ceramics, specifically two optimal compositions, BCZT0.85 (Ba_0.85_Ca_0.15_Zr_0.1_Ti_0.9_O_3_) and BCZT0.9 (Ba_0.9_Ca_0.1_Zr_0.1_Ti_0.9_O_3_), via the solid-state reaction method for the dielectric layer. The BCZT powders were calcined at varying temperatures (1200 and 1250 °C) and dwell times (2 and 4 h). The resulting phase formation and microstructure characteristics were analyzed using X-ray diffraction and scanning electron microscopy, respectively. Our investigation aimed to establish a correlation between the dielectric behavior and optical properties to determine the optimal composition and conditions for utilizing BCZT as thermal detectors in electric vehicle battery packs. All BCZT powders exhibited a tetragonal phase, as confirmed by JCPDS No. 01-079-2265. We observed an increase in the dielectric constant with higher calcining temperatures or longer dwell times. Remarkably, BCZT0.85 ceramic sintered at 1250 °C for 4 h displayed the highest dielectric constant of 15,342, establishing this condition as optimal for preparing the dielectric film with a maximum dielectric constant of 42. Furthermore, we investigated the temperature-dependent electroluminescence intensity of the samples, revealing a significant enhancement with increasing temperature, reaching its peak at 80 °C. Additionally, we observed a positive correlation between electroluminescence intensity and dielectric constant, indicating the potential for improved opto-thermal sensors. The findings from this study offer promising opportunities for the development of advanced opto-thermal sensors with potential applications in electric vehicle battery packs. Our work contributes to the expanding field of photoluminescent and electroluminescent systems by providing novel insights into the design and optimization of efficient and reliable sensors for thermal monitoring in electric vehicle technologies.

## 1. Introduction

Chemical processes have been traditionally used to synthesize classic photoluminescent (PL) and electroluminescent (EL) materials. These materials exist as sulfide, oxide, oxysulfide, nitride, and selenide compounds [1,2,3,4,5,6]. Using these powder precursors, thick PL and EL films are produced by screen printing or similar methods [1,7,8], comprising a highly efficient fluorescent system. Devices used for high-quality detection and inspection applications are subject to unwanted light scattering. Highly transparent thin film electroluminescent devices with a dielectric layer have optimal luminescent properties [9,10,11]. Luminescence can be achieved via the excitation of photons (e.g., fluorescence), electrons (e.g., cathodoluminescence), and ions (e.g., ionoluminescence), or the fabrication of electroluminescent systems [5,12,13,14,15,16]. In recent years, research focusing on systems made with inorganic materials has gained momentum due to their meager cost and availability. The performance of inorganic fluorescent films is temperature-dependent. Hence, this type of film has been used successfully in electric vehicle battery applications in the temperature range of 25–80 °C. The films can also withstand vibration. Due to these features, using inorganic films has several advantages, including a reduction in the number of wires, saving electricity, and faster temperature checks of electric batteries during operation. Furthermore, future work will involve the study of blending film simulation, examining variations in temperature at both the nano and macro scales [17,18,19].

The general design of light-emitting devices relies on thin films of phosphor sulfide incorporated with transition metals [2,20,21]. While piezoelectric materials such as BaTiO_3_, Pb(ZrTi)O_3_, PbTiO_3_, Ba(ZrTi)O_3_, and KNaNbO_3_ are widely used in the electronic industry [22], research efforts have been directed towards finding lead-free alternatives with comparable or superior properties. Barium-calcium zirconium titanate (BCZT) has emerged as a promising candidate, possessing enhanced dielectric and piezoelectric properties that rival those of lead-based materials [23,24,25]. Previous studies, such as our Previous work [26], have explored the preparation and characterization of BCZT ceramics using seed-induced methods. For instance, the incorporation of BaTiO_3_ (BT) seeds resulted in BCZT ceramics with a single perovskite phase and improved dielectric and piezoelectric characteristics. Additionally, Intatha and co-workers [27] investigated the impact of heterogeneous seed crystals, exemplified by SrFe_0.5_Nb_0.5_O_3_ (SFN), on the electrical properties of BCZT ceramics. The addition of SFN seeds influenced the phase behavior, dielectric properties, and polarization characteristics of the BCZT ceramics. These findings demonstrate the potential of BCZT ceramics as a lead-free dielectric material for use in light-emitting devices, including electroluminescent devices, where their improved emission efficiency and flexible nature make them well-suited for illumination-related applications [28].

This study aimed to investigate the fabrication and properties of electroluminescent devices. The initial step involved the preparation of barium-calcium zirconium titanate (BCZT) ceramic powder, which was utilized as the dielectric layer. Two morphotropic phase boundary (MPB) compositions, BCZT0.85 (Ba_0.85_Ca_0.15_Zr_0.1_Ti_0.9_O_3_) and BCZT0.9 (Ba_0.9_Ca_0.1_Zr_0.1_Ti_0.9_O_3_), were synthesized using the solid-state reaction method. Calcination of the BCZT powders was carried out at temperatures of 1200 and 1250 °C, with dwell times of 2 and 4 h, respectively. Subsequently, a four-layer film structure was constructed, consisting of an electrode layer, a dielectric layer, a phosphor layer, and a translucent conductive layer. The dielectric value of the dielectric layer was adjusted to facilitate electroluminescence of the cations in the phosphor layer through the activation mechanism of host-lattice energy transfer [29,30,31,32]. Modifying the dielectric value based on temperature variations resulted in corresponding changes in fluorescence, enabling the development of a compact and efficient opto-thermal sensor.

## 2. Materials and Methods

### 2.1. Synthesis and Characterization of BCZT

The Ba_0.85_Ca_0.15_Zr_0.1_Ti_0.9_O_3_ and Ba_0.9_Ca_0.1_Zr_0.1_Ti_0.9_O_3_ ceramic powders were prepared by using the solid-state reaction method. The starting materials were BaCO_3_ (99%, Sigma-Aldrich, St. Louis, MO, USA), CaCO_3_ (98.5–100.5%, Sigma-Aldrich), ZrO_2_ (99%, Sigma-Aldrich), and TiO_2_ (99%, Sigma-Aldrich) oxide powders. The oxide powders were weighed according to stoichiometric formulae, mixed using ethanol as a solvent, and ball-milled for 24 h. The slurry was dried at 100 °C for 24 h and sieved. The mixed compounds were calcined at different temperatures and dwell times at 1200 and 1250 °C for 2 and 4 h, respectively. Subsequently, the powder was uniaxially pressed into green pellets 10 mm in diameter and 1 mm in thickness at a one-ton compression. Finally, the samples were sintered at 1450 °C for 4 h. The phase formation of ceramic powder was observed by X-ray diffraction (XRD). Scanning electron microscopy (SEM) and the intercept method were used to examine the microstructure and grain size, respectively. To characterize the dielectric and ferroelectric properties, the surfaces of the sintered samples were polished and coated with silver paste on both sides to serve as electrodes.

### 2.2. Preparation and Characterization of EL Film

The electroluminescent films in this study were composed of four main layers: an electrode layer, a dielectric layer, a phosphor layer, and a translucent conductive layer, as illustrated in Figure 1. The films were prepared by using the spin coating method. In the first layer, the dielectric film was prepared by mixing BCZT powder with alkyd resin binder, deposited on a copper sheet, and spread out. In the phosphor layer, ZnS powder was mixed with binder and deposited over the dielectric layer. Finally, the translucent conductive layer was prepared by spin coating over the phosphor layer. Each layer was individually dried at room temperature for 30 min before coating the next layer.

The BCZT0.85 and BCZT0.9 were synthesized from Ba_0.85_Ca_0.15_Zr_0.1_Ti_0.9_O_3_ and Ba_0.9_Ca_0.1_Zr_0.1_Ti_0.9_O_3_ systems, respectively, and each condition was calcined at different temperatures and dwell times at 1200 and 1250 °C for 2 and 4 h, respectively (Table 1).

## 3. Results and Discussion

### 3.1. XRD Phase Analysis of BCZT Powders

In this work, BCZT ceramics were synthesized using the solid-state reaction method. The phase formation of the BCZT powder was analyzed using XRD (Figure 2 and Figure 3). The samples were calcined at different temperatures and dwell times: 1200 °C for 2 h, 1200 °C for 4 h, 1250 °C for 2 h, and 1250 °C for 4 h. Based on the XRD patterns for 2θ = 29, the sample calcined at 1200 °C for 2 h had an unidentified phase. However, upon increasing the calcining temperature or dwell time to 1200 °C for 4 h, 1250 °C for 2 h, and 1250 °C for 4 h, the unidentified phase disappeared. The resulting XRD patterns had a pure perovskite without a secondary peak of impurity [30,31]. Regarding the expanded XRD patterns for 2θ = 36–49°, the split of the single peaks of (111) reflections was observed at 2θ~38° to 40° and the single peak of (002) reflection at 2θ~44° to 46° in all samples used in this study [33]. This suggested that all the samples had a pseudo-cubic phase [34].

### 3.2. Microstructure Analysis of BCZT Ceramics

The SEM micrographs in Figure 4 and Figure 5 present the sintered surfaces of BCZT0.85 and BCZT0.9 ceramics at 1450 °C for 4 h. It was observed that the grain morphology remained relatively unchanged in both ceramics. The grains exhibited irregular polyhedral shapes and a non-homogeneous distribution. Notably, the microstructure of BCZT ceramics was significantly influenced by the calcining temperature. Specifically, increasing the calcining temperature and dwell time led to an increase in grain size. Calcining at 1200 °C for 2 h resulted in insufficient crystal formation, leading to smaller grain sizes. Conversely, higher temperatures and longer dwell times resulted in a larger grain size distribution, as confirmed by SEM analysis. Furthermore, a correlation was observed between grain size and light intensity (LV intensity); as the grain size increased, the light intensity also increased. This finding is consistent with the results obtained from the light intensity test. Figure 6 showcases an SEM image of the BCZT and ZnS surfaces, along with a cross-section micrograph of a sandwiched electroluminescent (EL) device. The image illustrates the significantly larger particle size of ZnS in comparison to BCZT. The cross-section micrograph highlights the uniform microstructure of the entire EL device, comprising a copper sheet, a BCZT layer, and a ZnS layer. The micrograph provides clear visibility of the stacking order and emphasizes the compactness of the film microstructure, as well as the planar interfaces between the phosphor and dielectric layers. These microstructural features contribute to the high optical quality of the entire structure, as evident from the electroluminescent spectra displayed in Figures 16 and 17, which demonstrate visible spectrum emission.

This explanation provides a clear understanding of the results by emphasizing the relationship between calcination temperature, dwell time, and grain size in BCZT ceramics. Furthermore, it underscores the significance of the observed correlation between grain size and light intensity. Additionally, the discussion highlights the microstructural features of the EL device, confirming its uniformity and supporting the visible spectrum emission observed in the electroluminescent spectra.

### 3.3. Electrical Properties of BCZT Ceramics

The dielectric constant (εr) and dielectric loss (tanδ) of ceramic samples measured at room temperature as a function of frequency are displayed in Figure 7 and Figure 8. The dielectric constant and dielectric loss of the Ba_0.85_Ca_0.15_Zr_0.1_Ti_0.9_O_3_ ceramics calcined at different temperatures and dwell times were compared. The maximum dielectric constant of the BCZT0.85_1250_2h sample at 1 kHz was 3326, while that of the BCZT0.85_1250_4h sample was 2910 (Figure 7). The maximum dielectric constant at 1 kHz of the BCZT0.9_1250_4h sample was 2905, while the minimum dielectric constant at 1 kHz was 2368 for the BCZT0.9_1200_4h sample. Overall, the dielectric constant of all Ba_0.85_Ca_0.15_Zr_0.1_Ti_0.9_O_3_ ceramic samples was higher than all Ba_0.9_Ca_0.1_Zr_0.1_Ti_0.9_O_3_ ceramic samples. The temperature-related dielectric shift impacts light intensity, which is helpful for measuring light intensity versus temperature.

The relationship between dielectric properties and temperature from 30 °C to 250 °C for Ba_0.85_Ca_0.15_Zr_0.1_Ti_0.9_O_3_ and Ba_0.9_Ca_0.1_Zr_0.1_Ti_0.9_O_3_ is shown in Figure 9 and Figure 10, respectively. The maximum dielectric peak of all ceramics exhibited only a slight variation with frequency. The maximum dielectric constant (ε_r_) of Ba_0.85_Ca_0.15_Zr_0.1_Ti_0.9_O_3_ and Ba_0.9_Ca_0.1_Zr_0.1_Ti_0.9_O_3_ ceramics at Curie temperature increased modestly with increasing calcine temperature and dwell time, ranging from 11,314–15,341 and 8828.78–13,455.9 (measured at 1 kHz), respectively. Dielectric loss (tanδ) showed similar behavior to that of the dielectric constant. No significant change was observed with increasing frequency, calcine temperature, or dwell time. The lowest tanδ value of all samples measured between 30 and 250 °C was 0.01807. Hence, the calcining temperature and dwell time had an effect on the dielectric loss of BCZT ceramics. Furthermore, the Curie temperature (TC) of ceramics, which ranged from 95 to 100 °C, did not significantly change with increased calcine temperature and dwell time.

The hysteresis loop confirmed the ferroelectric properties of the BCZT ceramics. The loops between polarization (P) and an applied electric field (E) showed saturation polarization when an electric field of ∼5 kV/mm was used (Figure 11a). Plots of the ferroelectric parameters such as remnant polarization (Pr) and coercive field (Ec) as a function of different calcine temperatures and dwell times of Ba_0.85_Ca_0.15_Zr_0.1_Ti_0.9_O_3_ and Ba_0.9_Ca_0.1_Zr_0.1_Ti_0.9_O_3_ ceramics are shown in Figure 11b. The calcine temperature and dwell time had no significant effect on the Pr and Ec values. The highest Pr value was found in the sample BCZT0.9_1250_4 h, while an Ec value of 3.4921 kV/cm was obtained for the BCZT0.85_1250_4 h sample. The observed hysteresis loop in the ferroelectric properties of BCZT ceramics confirms their ferroelectric nature, indicating their ability to switch polarization in response to an applied electric field. While the specific influence of these ferroelectric properties on the dielectric properties of the electroluminescent film is not explicitly addressed, it is worth considering the potential indirect relationship between these properties. The dielectric properties of a material reflect its ability to store and release electrical energy in response to an electric field. In the case of electroluminescent film, changes in the dielectric properties can impact the overall efficiency of energy transfer and light emission. Therefore, variations in the dielectric properties, influenced by factors like temperature and calcine time, as investigated in the study, may indirectly affect the luminescence intensity of the electroluminescent film. It is important to note that the specific quantitative analysis and precise understanding of this relationship require further investigation. Additional experiments and measurements would be necessary to establish the direct correlation between the ferroelectric and dielectric properties in the context of the electroluminescent film. These future studies could involve systematically varying the calcine temperature and dwell time, measuring the resulting ferroelectric and dielectric properties, and assessing their impact on the luminescence intensity of the film.

Furthermore, it is essential to acknowledge the limitations and potential confounding factors in the experimental design that could influence the observed relationship. Factors such as film thickness, composition, and microstructure could also play a role in the overall performance of the electroluminescent film and its relationship with the ferroelectric and dielectric properties.

### 3.4. Electrical Properties of BCZT Films

To characterize the electrical properties of BCZT film, a sample was prepared by spinning the BCZT coating powder with a binder on copper sheet. Subsequently, a silver electrode was coated with a BCZT layer. The εr and tanδ at room temperature as a function of frequency are displayed in Figure 12 and Figure 13. Both calcine temperature and dwell time affect the dielectric properties of BCZT ceramics. The εr value increased with increasing calcine temperature and dwell time. The εr values of film BCZT0.85 samples were in the range of 23.064–29.672, while those of film BCZT0.9 samples ranged from 23.618–32.171. The BCZT0.85_1250_2 h sample had the highest tanδ at 1 kHz. Other samples showed tanδ less than 0.018. In general, the film BCZT0.9 sample had a higher dielectric constant than the film BCZT0.85. This could be a result of the calcined temperature, which is associated with changes in the microstructure of the ceramic.

The temperature dependence of the dielectric constant and dielectric loss at a frequency of 10 kHz is presented in Figure 14 and Figure 15. The dielectric constants of BCZT0.85 and BCZT0.9 were compared using the graph with calcined temperatures and dwell times of 1200 °C at 2–4 h and 1250 °C at 2–4 h, respectively. Both the calcine temperature and dwell time had a significant effect on the dielectric constant. The dielectric constant at Curie temperature (TC) increased with increasing calcine temperature and dwell time. The maximum value was 42, which was the calcined temperature at 1250 °C and dwelling at 4 h for the BCZT0.9 sample. The TC of the film BCZT samples was in the range of 120–147 °C.

### 3.5. Electroluminescent Properties

Figure 16 and Figure 17 present the electroluminescent spectra of the EL devices with various calcine temperatures and dwell times of BCZT0.85 or BCZT0.9 on the dielectric layer. The blue electroluminescence emission from all devices mentioned above had a peak intensity at a wavelength of 499–506 nm and exhibited a full width half maximum (FWHM) of 83–84 nm when heated at 80 °C. By comparison, the light intensity of BCZT in the dielectric layer, along with the electroluminescence intensity, increased when the calcining temperature and dwell time increased. The electroluminescence intensity of the EL devices with BCZT0.85 coating on the dielectric layer ranged from 2611 to 3812 W/m^2^, whereas that of the dielectric layer coated with BCZT0.9 was between 2178 and 3358 W/m^2^. Moreover, the electroluminescence intensity increased in a temperature-dependent manner. Hence, manipulating temperature and dielectric constant leads to improved fluorescence, which could be efficiently used in the field of opto-thermal sensors.

Figure 18 illustrates a linear relationship between intensity and temperature. Our laboratory-developed device emits blue light at approximately 503 nm. The Commission Internationale de L’éclairage (CIE) coordinates at 25 °C are (x = 0.1538, y = 0.2015). At 80 °C, the CIE coordinates shift to (x = 0.1640, y = 0.2724) (refer to Figure 19). Additionally, Figure 20 presents a photographic image of the EL device featuring BCZT0.85_1250_4 h on the dielectric layer.

## 4. Conclusions

In this work, we have applied a variation of the spine coating procedure to prepare electroluminescent films consisting of four main layers: an electrode layer, a dielectric layer, a phosphor layer, and a translucent conductive layer. The Ba_0.85_Ca_0.15_Zr_0.1_Ti_0.9_O_3_ and Ba_0.9_Ca_0.1_Zr_0.1_Ti_0.9_O_3_ ceramics were exposed to different calcine temperatures and dwell times on the dielectric layer of electroluminescent films. This methodology has provided evidence that different calcine temperatures and dwell times of dielectric powder have an effect on the ceramic microstructure and dielectric properties of ceramics and films, besides affecting the intensity of luminescent emission. All of the BCZT powders exhibited the tetragonal phase, which was confirmed by JCPDS No. 01-079-2265, and the particle sizes of BCZT increased with increasing calcination time from 2 to 4 h. The dielectric constant (εr) at Curie temperature of BCZT ceramics and films tends to increase modestly with increasing calcine temperature and dwell time. The electroluminescence intensity increased when the calcine temperature and dwell duration rose. Moreover, the EL intensity value tended to rise as the temperature rose. This remarkable performance demonstrates the feasibility of using the electroluminescent film as Ba_0.85_Ca_0.15_Zr_0.1_Ti_0.9_O_3_ calcine at 1250 °C for 4 h on the dielectric layer using the spin coating technique for the entire fabrication of electroluminescent multilayer devices.

## Figures and Tables

**Figure 1 materials-16-05202-f001:**
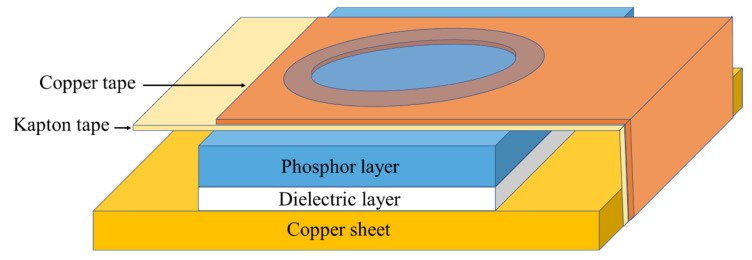
Scheme of the electroluminescence device.

**Figure 2 materials-16-05202-f002:**
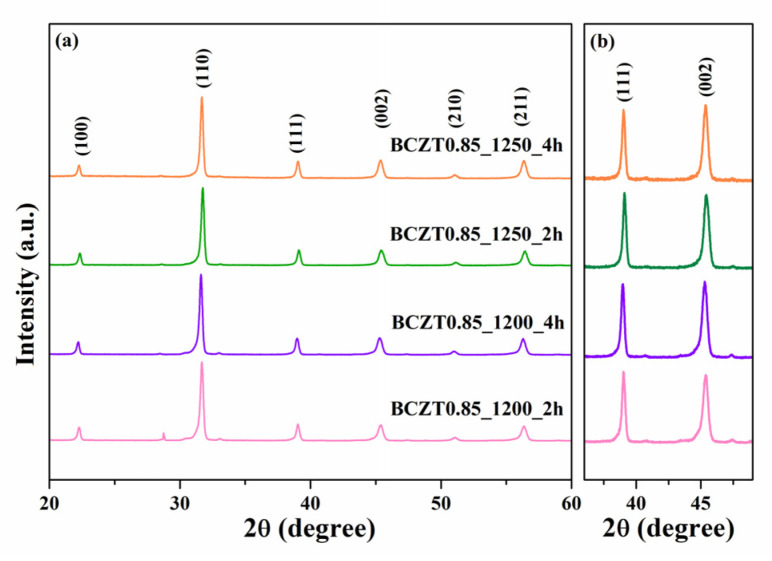
X-ray diffraction patterns of Ba_0.85_Ca_0.15_Zr_0.1_Ti_0.9_O_3_: (**a**) XRD patterns for 2θ = 20°–60° (**b**) the expanded XRD patterns for 2θ = 36°–49°.

**Figure 3 materials-16-05202-f003:**
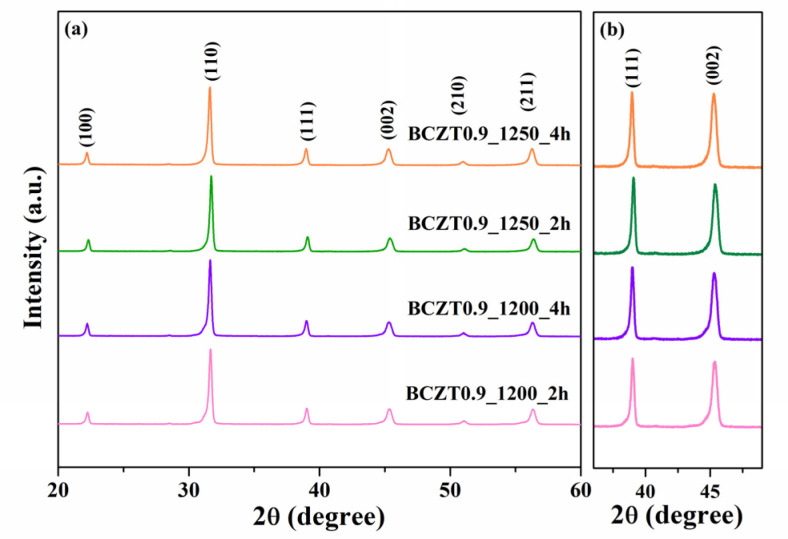
X-ray diffraction patterns of Ba_0.9_Ca_0.1_Zr_0.1_Ti_0.9_O_3_: (**a**) XRD patterns for 2θ = 20°–60° (**b**) the expanded XRD patterns for 2θ = 36°–49°.

**Figure 4 materials-16-05202-f004:**
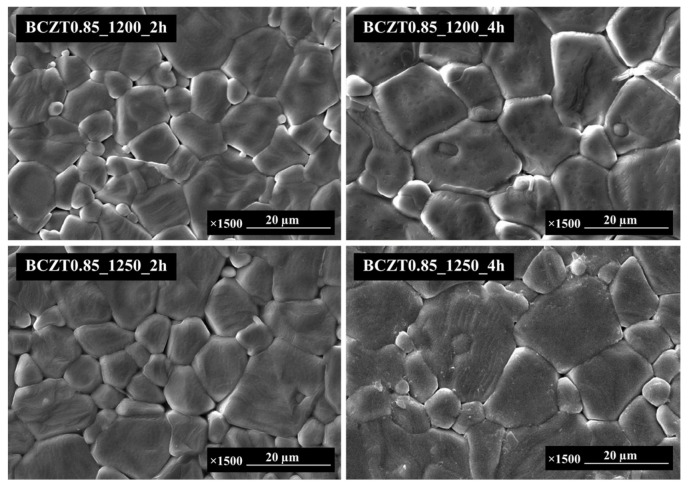
SEM micrographs of the surface Ba_0.85_Ca_0.15_Zr_0.1_Ti_0.9_O_3_.

**Figure 5 materials-16-05202-f005:**
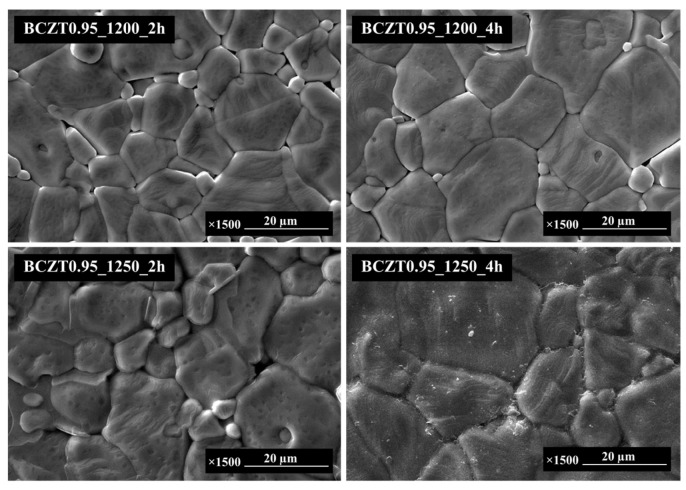
SEM micrographs of the surface Ba_0.9_Ca_0.1_Zr_0.1_Ti_0.9_O_3_.

**Figure 6 materials-16-05202-f006:**
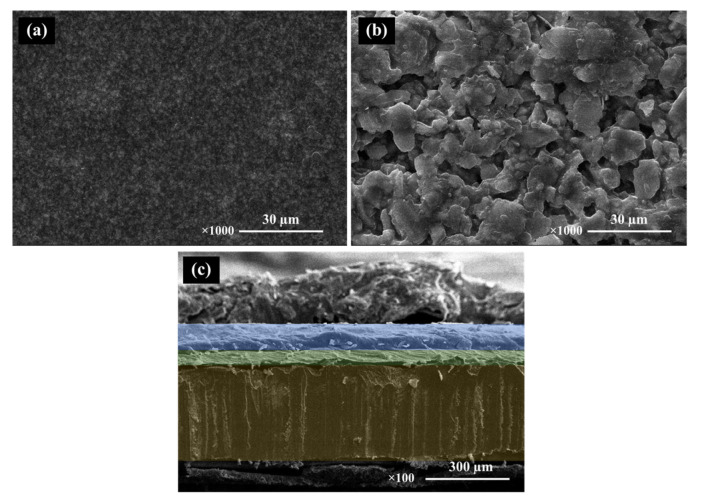
SEM micrographs of the surface BCZT film (**a**), ZnS film (**b**), and SEM micrographs of the cross-section of the multilayer structure of the EL device (**c**).

**Figure 7 materials-16-05202-f007:**
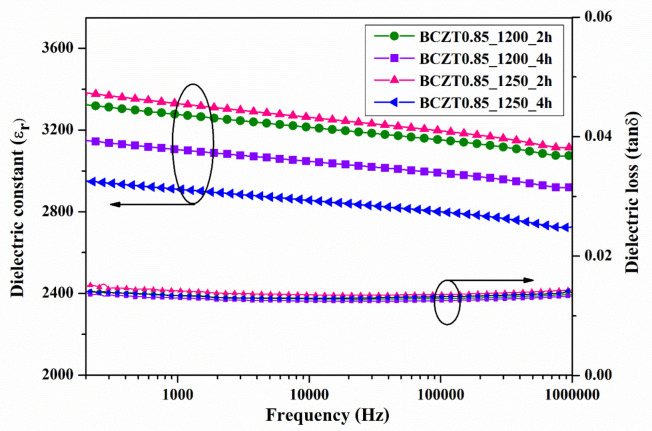
Dielectric constant (εr) and dielectric loss (tanδ) of Ba_0.85_Ca_0.15_Zr_0.1_Ti_0.9_O_3_ ceramics as a function of frequency at room temperature.

**Figure 8 materials-16-05202-f008:**
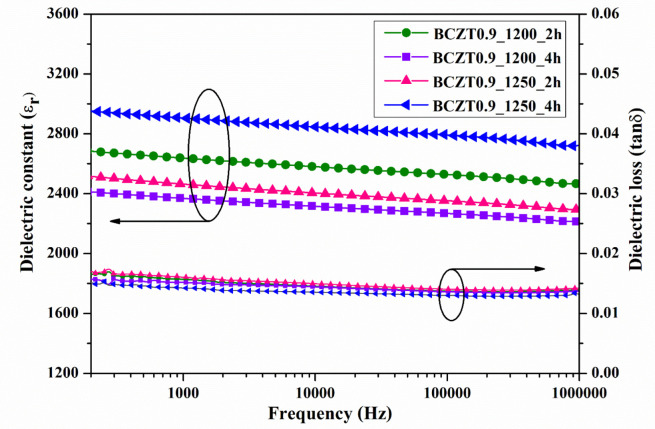
Dielectric constant (εr) and dielectric loss (tan δ) of Ba_0.9_Ca_0.1_Zr_0.1_Ti_0.9_O_3_ ceramics as a function of frequency at room temperature.

**Figure 9 materials-16-05202-f009:**
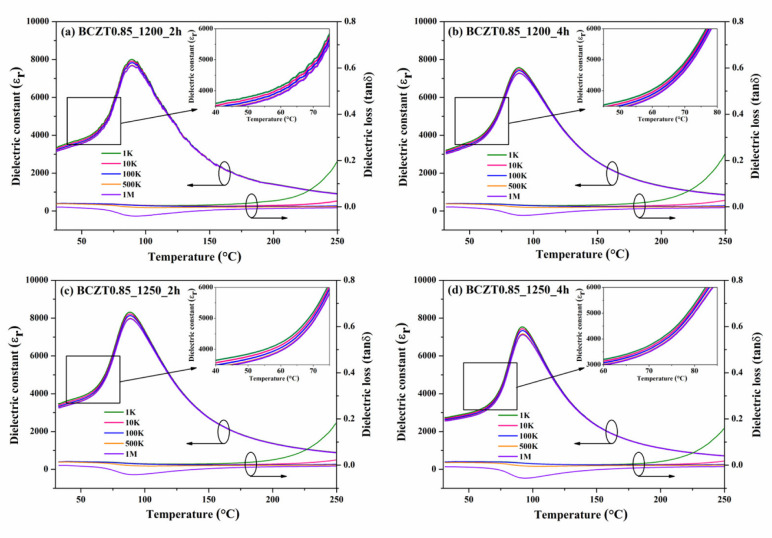
Temperature dependence of the dielectric constant of Ba_0.85_Ca_0.15_Zr_0.1_Ti_0.9_O_3_ ceramics: (**a**) BCZT0.85_1200_2 h, (**b**) BCZT0.85_1200_4 h, (**c**) BCZT0.85_1250_2 h, and (**d**) BCZT0.85_1250_4 h.

**Figure 10 materials-16-05202-f010:**
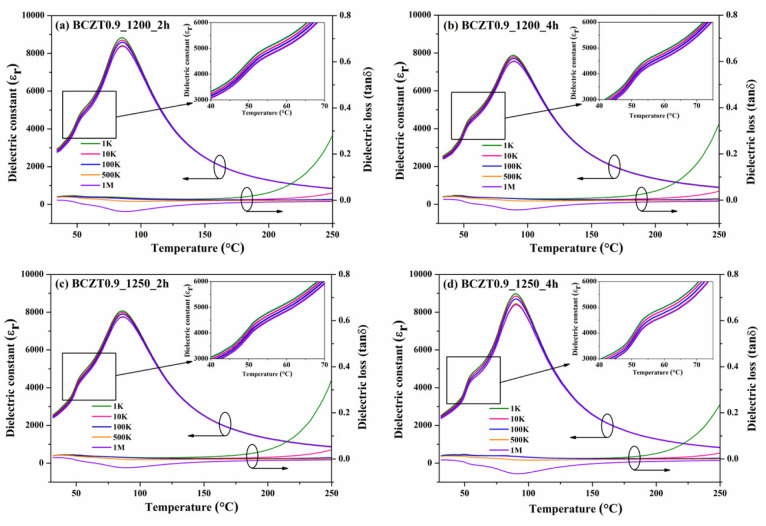
Temperature dependence of the dielectric constant of Ba_0.85_Ca_0.15_Zr_0.1_Ti_0.9_O_3_ ceramics: (**a**) BCZT0.9_1200_2 h, (**b**) BCZT0.9_1200_4 h, (**c**) BCZT0.9_1250_2 h, and (**d**) BCZT0.9_1250_4 h.

**Figure 11 materials-16-05202-f011:**
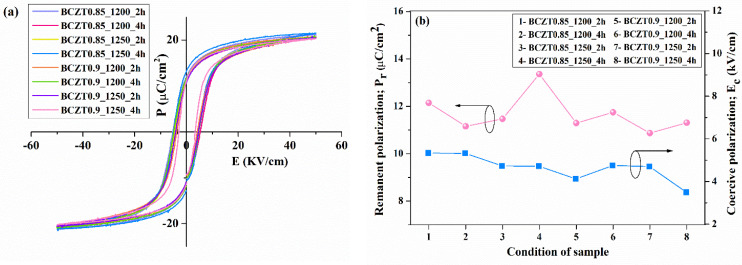
(**a**) Ferroelectric hysteresis loops and (**b**) the hysteresis parameter of BCZT.

**Figure 12 materials-16-05202-f012:**
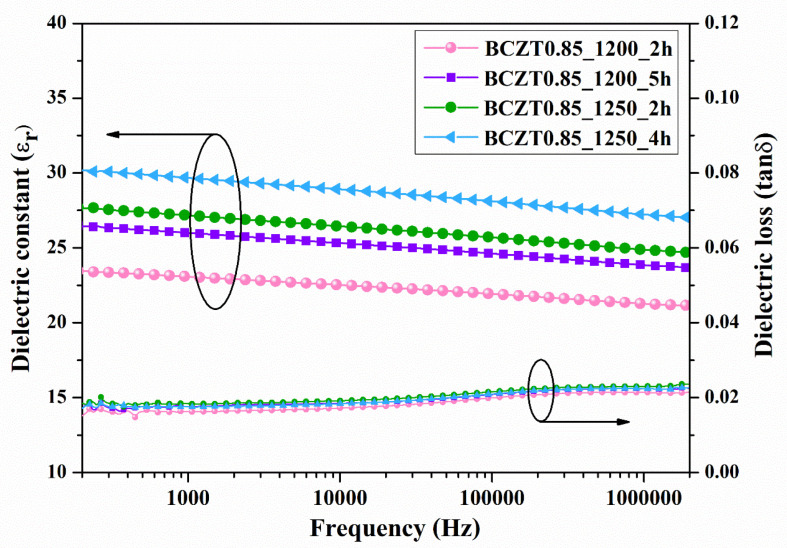
Dielectric constant (εr) and dielectric loss (tanδ) of Ba_0.85_Ca_0.15_Zr_0.1_Ti_0.9_O_3_ films as a function of frequency at room temperature.

**Figure 13 materials-16-05202-f013:**
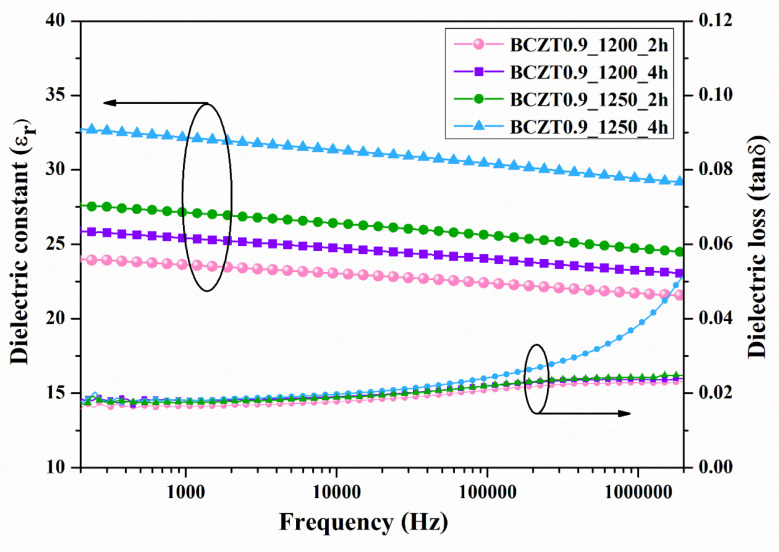
Dielectric constant (εr) and dielectric loss (tanδ) of Ba_0.9_Ca_0.1_Zr_0.1_Ti_0.9_O_3_ films as a function of frequency at room temperature.

**Figure 14 materials-16-05202-f014:**
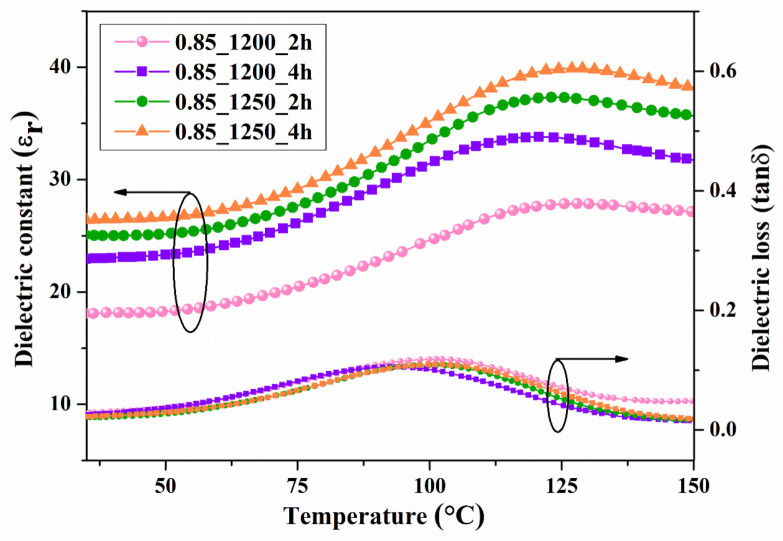
Temperature dependence of the dielectric constant of Ba_0.85_Ca_0.15_Zr_0.1_Ti_0.9_O_3_ films.

**Figure 15 materials-16-05202-f015:**
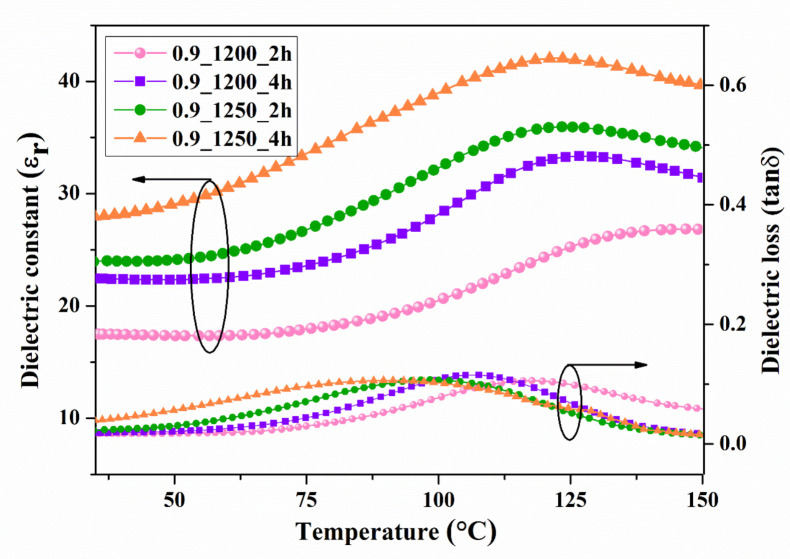
Temperature dependence of the dielectric constant of Ba_0.9_Ca_0.1_Zr_0.1_Ti_0.9_O_3_ films.

**Figure 16 materials-16-05202-f016:**
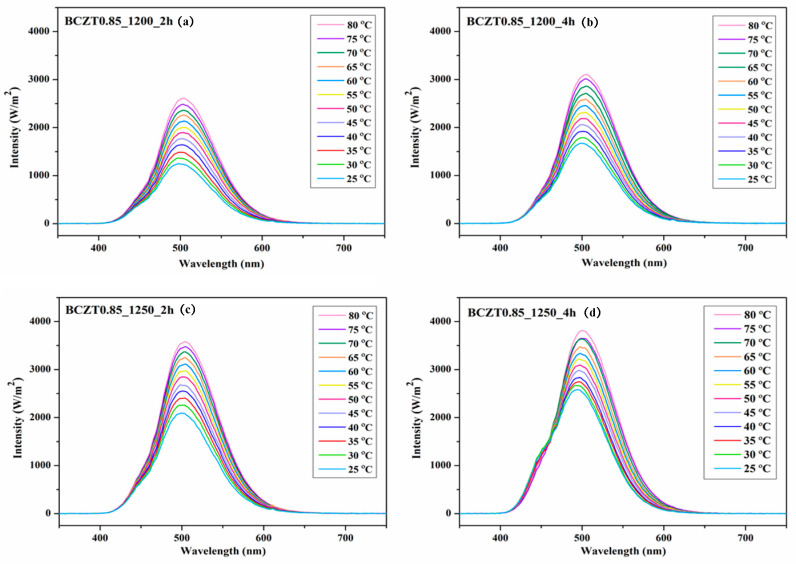
Electroluminescent (EL) spectra for the devices with different calcine temperatures and dwell times of Ba_0.85_Ca_0.15_Zr_0.1_Ti_0.9_O_3_ on the dielectric layer: (**a**) BCZT0.85_1200_2 h, (**b**) BCZT0.85_1200_4 h, (**c**) BCZT0.85_1250_2 h, and (**d**) BCZT0.85_1250_4 h.

**Figure 17 materials-16-05202-f017:**
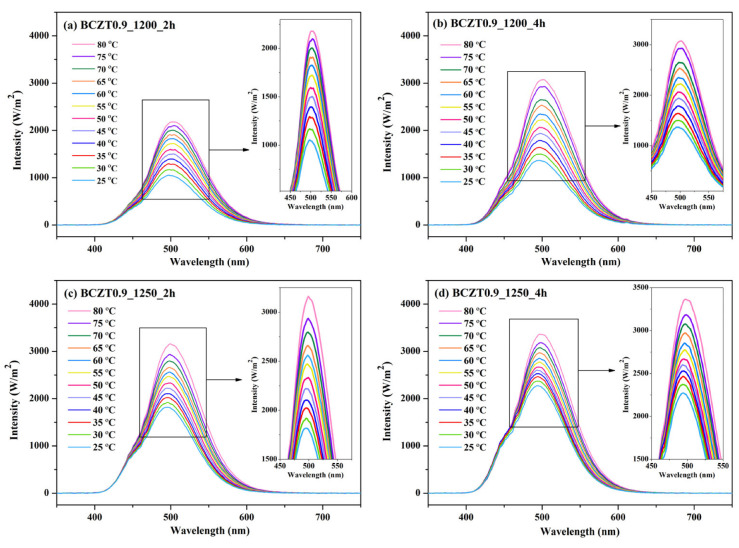
Electroluminescent (EL) spectra for the devices with different calcine temperatures and dwell times of Ba_0.9_Ca_0.1_Zr_0.1_Ti_0.9_O_3_ on the dielectric layer: (**a**) BCZT0.9_1200_2 h, (**b**) BCZT0.9_1200_4 h, (**c**) BCZT0.9_1250_2 h, and (**d**) BCZT0.9_1250_4 h.

**Figure 18 materials-16-05202-f018:**
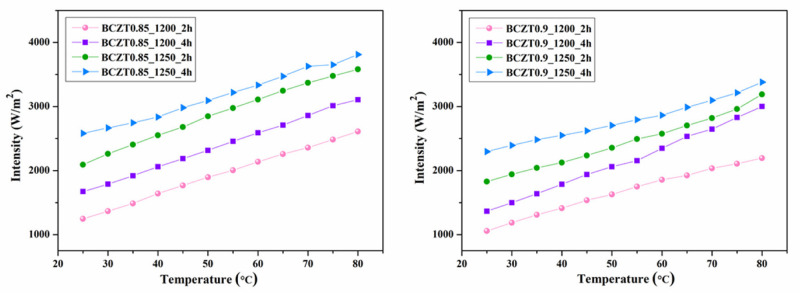
EL intensity and temperature of electroluminescent samples.

**Figure 19 materials-16-05202-f019:**
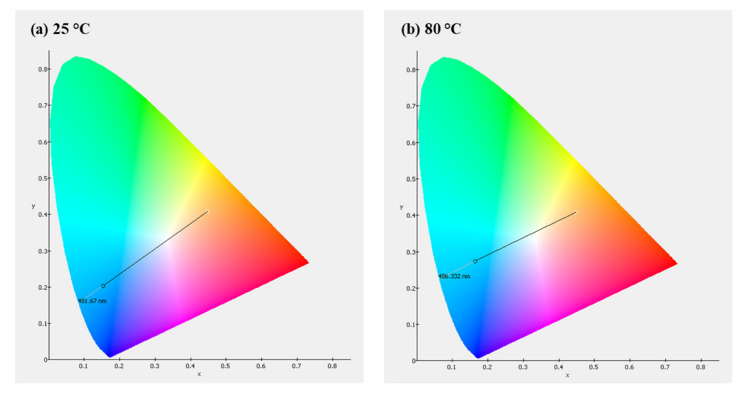
The CIE color coordinates of an EL device: (**a**) at temperature 25 °C and (**b**) at temperature 80 °C.

**Figure 20 materials-16-05202-f020:**
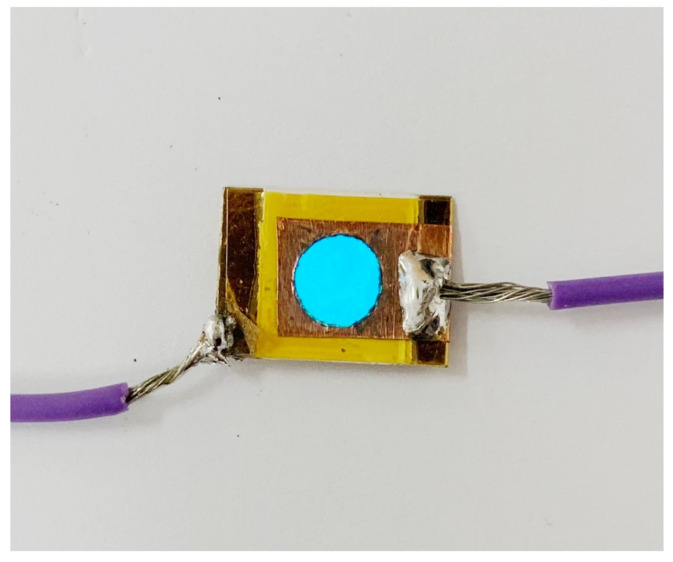
Photo image of the EL device constructed in this study.

**Table 1 materials-16-05202-t001:** The conditions of BCZT ceramics used in this study.

Sample	Title 2	Title 3
	Temperature (°C)	Dwell Time (h)
BCZT0.85_1200_2 h	1200	2
BCZT0.85_1200_4 h	1200	4
BCZT0.9_1200_2 h	1250	2
BCZT0.9_1200_4 h	1250	4

## Data Availability

Not applicable.

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
