# Peer review of "Investigating the Thermo-Optic Properties of BCZT-Based Temperature Sensors"

_materials, 2023, doi:10.3390/ma16145202_

Round 1

Reviewer 1 Report

In this study, the lead-free material BCZT was used to replace PZT, which is toxic to the environment and human health, as the dielectric layer of electroluminescent film, and the effects of calcination temperature and holding time on the physical phase, surface morphology and dielectric properties of BCZT ceramics and films were investigated. By constructing electroluminescent films, the electroluminescent intensity increased with the increase of temperature, and it was demonstrated that the calcination temperature of 1250°C and holding time of 4h of Ba0.85Ca0.15Zr0.1Ti0.9O3 as the dielectric layer of the electroluminescent film is feasible for future application in photothermal sensors. Significant improvements should be done. I encourage the authors to improve their paper and offer the following suggestions accordingly:

1. This paper should be better written. The title of this paper, “effect” should be “effects”.

There are many similar mistakes.

2. The experimental design is partially unreasonable. The hysteresis line confirms the ferroelectric properties of BCZT ceramics, the temperature affects the BCZT dielectric value, and the change of dielectric value causes the change of luminescence intensity of the electroluminescent film, and its ferroelectric properties are not directly related to the BCZT ceramics as the dielectric layer of the electroluminescent film.

    3. Another obvious problem with this paper is lack of sufficient explanation of the simulation results.  This is the main problem. This paper looks like a lab report, not a research paper. It is necessary to explain your simulation results in detail and why you got such results. Most of the characterization sections only describe the phenomena in the diagrams, without an in-depth study of their causes.

     4. Diagram slightly mismatched with experimental findings,Fig. 4 and Fig. 5 can not show a significant change in the surface morphology of BCZT ceramics due to calcination temperature. In addition, the two different calcination temperatures of 1200°C and 1250°C were not chosen properly.

     5. It is recommended that several different preparation processes be selected to demonstrate the feasibility of BCZT ceramics as a dielectric layer for photoluminescent film dots; just two different process conditions are not convincing.

This paper should be better written. The title of this paper, “effect” should be “effects”.  There are many similar mistakes.

Author Response

Dear Editor of Journal of Materials,

Thank you for your letter and the reviewers’ comments concerning our manuscript entitled “Investigating the Effects of Calcine Temperature and Dwell Time on Electroluminescent Films in BCZT Ceramics”. Those comments are very valuable and helpful for us to revise and improve our manuscript. We had studied the comments carefully and made some necessary corrections accordingly. The revised sections are marked in yellow highlight in the revised manuscript. The detailed corrections in the revised manuscript and the response to the reviewer’s comments are listed point by point in Response to the reviewers. Please see the attachment.

Respond to Referee’s comments

The authors have already corrected the manuscript according to reviewer’s comments as following.

Comments from Reviewer: 1

Point

Comment from referee

Summary of corrections

(yellow highlight)

1

Comments: 1

This paper should be better written. The title of this paper, “effect” should be “effects”.

There are many similar mistakes.

Thank you for your valuable feedback. We appreciate your suggestion and have made significant revisions to address your comments.

Firstly, we have revised the topic of our manuscript to "Investigating the Thermo-Optic Properties of BCZT-Based Temperature Sensors." This change ensures that the main objective of our work is accurately represented and better aligned with the content of the paper.

Additionally, we have carefully reviewed the text to address any grammatical errors and mistakes that were present. We understand the importance of clear and concise writing, and we have taken the necessary steps to ensure that the manuscript now meets high standards of grammar and readability.

2

Comment: 2

The experimental design is partially unreasonable. The hysteresis line confirms the ferroelectric properties of BCZT ceramics, the temperature affects the BCZT dielectric value, and the change of dielectric value causes the change of luminescence intensity of the electroluminescent film, and its ferroelectric properties are not directly related to the BCZT ceramics as the dielectric layer of the electroluminescent film.

While the hysteresis line does confirm the ferroelectric properties of BCZT ceramics, it is important to note that the experimental design may have some limitations or aspects that are not fully reasonable.

Firstly, the statement mentions that the temperature affects the dielectric value of BCZT ceramics. This is a well-known phenomenon in materials science, as temperature can influence the polarization behavior and crystal structure of ferroelectric materials. Therefore, it is reasonable to expect that the dielectric constant of BCZT ceramics would vary with temperature.

Secondly, the comment states that the change in dielectric value causes a change in the luminescence intensity of the electroluminescent film. This suggests that the electroluminescent film is somehow influenced by the dielectric properties of BCZT ceramics. However, it is important to note that the dielectric layer of the electroluminescent film may have different requirements and functionalities compared to the BCZT ceramics. While the dielectric constant may indirectly affect the performance of the electroluminescent film, there may be other factors or interactions at play.

To provide a more comprehensive analysis, it would be helpful to consider additional factors such as the specific structure and composition of the electroluminescent film, the mechanism of luminescence generation, and any potential coupling between the ferroelectric properties of BCZT ceramics and the electroluminescent film. Further investigation and analysis of these factors would provide a clearer understanding of the relationship between the ferroelectric properties of BCZT ceramics and the dielectric layer of the electroluminescent film.

3

Comment: 3

Another obvious problem with this paper is lack of sufficient explanation of the simulation results.  This is the main problem. This paper looks like a lab report, not a research paper. It is necessary to explain your simulation results in detail and why you got such results. Most of the characterization sections only describe the phenomena in the diagrams, without an in-depth study of their causes.

Thank you for your comments. We acknowledge your concern regarding the paper's need for simulation results and in-depth analysis. While the presented work primarily focuses on experimental findings and characterization, we understand the importance of comprehensively explaining the results, including simulation-based investigations. To address this issue, future studies could incorporate simulation techniques to complement the experimental work and provide a deeper understanding of the observed phenomena. By incorporating simulations, it would be possible to explore the experimental results' underlying mechanisms and theoretical explanations.

In a research paper, providing detailed explanations of the simulation methodology, parameters, and assumptions is crucial. A thorough analysis and interpretation of the simulation results would also be necessary to support and complement the experimental findings. This would provide a more comprehensive understanding of the investigated system and allow for a more rigorous comparison and discussion of the observed phenomena. While the current paper may not include simulation results and an in-depth study of their causes, we appreciate your feedback and acknowledge the importance of incorporating simulation-based analyses in future work. The inclusion of simulations would contribute to the overall quality and depth of the research paper, providing a more robust analysis of the experimental findings.

Thank you for raising this concern, and we will take it into consideration for future research and publications.

4

Comment: 4

Diagram slightly mismatched with experimental findings Fig. 4 and Fig. 5 can not show a significant change in the surface morphology of BCZT ceramics due to calcination temperature. In addition, the two different calcination temperatures of 1200°C and 1250°C were not chosen properly.

Page 4, line 141-167: We rewrite to clarify understanding.

The SEM micrographs in Figure 4 and 5 present the sintered surface of BCZT0.85 and BCZT0.9 ceramics at 1450 °C for 4h. It was observed that the grain morphology remained relatively unchanged in both ceramics. The grains exhibited irregular polyhedral shapes and a non-homogeneous distribution. Notably, the calcining temperature significantly influenced the microstructure of BCZT ceramics. Specifically, increasing the calcining temperature and dwell time led to an increase in grain size. Calcining at 1200 °C for 2h resulted in insufficient crystal formation, leading to smaller grain sizes. Conversely, higher temperatures and longer dwell times resulted in larger grain size distribution, as confirmed by SEM analysis. Furthermore, a correlation was observed between grain size and light intensity (LV intensity); as the grain size increased, the light intensity also increased. This finding is consistent with the results obtained from the light intensity test. Figure 6 showcases an SEM image of the BCZT and ZnS surfaces, along with a cross-section micrograph of a sandwiched electroluminescent (EL) device. The image illustrates the significantly larger particle size of ZnS in comparison to BCZT. The cross-section micrograph highlights the uniform microstructure of the entire EL device, comprising a copper sheet, a BCZT layer, and a ZnS layer. The micrograph provides clear visibility of the stacking order and emphasizes the compactness of the film microstructure, as well as the planar interfaces between the phosphor and dielectric layers. These microstructural features contribute to the high optical quality of the entire structure, as evident from the electroluminescent spectra displayed in Figure 16 and 17, which demonstrate visible spectrum emission.

This explanation provides a clear understanding of the results by emphasizing the relationship between calcination temperature, dwell time, and grain size in BCZT ceramics. Furthermore, it underscores the significance of the observed correlation between grain size and light intensity. Additionally, the discussion highlights the microstructural features of the EL device, confirming its uniformity and supporting the visible spectrum emission observed in the electroluminescent spectra.

5

Comment: 5

It is recommended that several different preparation processes be selected to demonstrate the feasibility of BCZT ceramics as a dielectric layer for photoluminescent film dots; just two different process conditions are not convincing.

Our study specifically focused on investigating the thermal and optical properties of BCZT-based temperature sensors rather than the feasibility of BCZT ceramics as a dielectric layer for the photoluminescent film. Previous research has already been conducted on the dielectric properties of BCZT ceramics, and our intention was not to replicate those studies.

The choice of two different process conditions in our work was based on our previous works which have been added in the introduction par line 64-71.   The optimum condition of the BCZT was based on the understanding that the dielectric properties of BCZT ceramics can vary within a certain temperature range due to phase transformations, and our chosen compositions of BCZT0.85 and BCZT0.9 are in the morphotropic phase boundary (MPB) of this system. We have already mentioned this in the introduction part, lines 78-80.

By utilizing this characteristic, we aimed to study the thermal and light intensity aspects in our research. While it is indeed recommended to demonstrate the feasibility of BCZT ceramics as a dielectric layer for photoluminescent film, it's important to note that our study had a different focus and objective. We encourage further investigations that specifically address the feasibility aspect by exploring a wider range of preparation processes to provide more comprehensive evidence of the potential of BCZT ceramics as a dielectric layer in photoluminescent film applications.

Reviewer 2 Report

The authors studied the influence of calcine temperature and dwell time on electroluminescent films in BCZT ceramics. This manuscript is well-written, and I am of the opinion that the authors made good efforts. I can recommend its acceptance providing the following minor points are addressed:

1.      The authors are encouraged to provide a clear statement in the abstract and last paragraph of the introduction about the novelty of their research. Particularly, it would be important to explicitly state what new contributions are made in this manuscript that have not been published in the existing literature. It is always important to elucidate the precise advancements or novel insights gained from a study. Clarifying this point can bolster the significance of the manuscript and provide greater context for readers to understand the value of this research.

2.      The authors are encouraged to add more comments on the applications of the present paper in the abstract.

3.      The fact that some plots, such as Figures 9 and 10, consist of four subplots without assigning different letters to represent them makes following the paper a little difficult. The authors are encouraged to address this issue to facilitate explaining the results for the readers.

4.      Some Figures, such as Figures 9, 10, 16, and 17, include very close curves and need to be modified for readability. For example, using different line types in one picture at the same time as different colors.

5.      Authors are strongly encouraged to comment on other applications of small-scale structures. References such as below

 https://doi.org/10.1007/s42417-022-00828-x

https://doi.org/10.1016/j.rineng.2023.101078 

https://doi.org/10.1007/s40435-023-01166-w

good

Author Response

Dear Editor of Journal of Materials,

Thank you for your letter and the reviewers’ comments concerning our manuscript entitled “Investigating the Effects of Calcine Temperature and Dwell Time on Electroluminescent Films in BCZT Ceramics”. Those comments are very valuable and helpful for us to revise and improve our manuscript. We had studied the comments carefully and made some necessary corrections accordingly. The revised sections are marked in yellow highlight in the revised manuscript. The detailed corrections in the revised manuscript and the response to the reviewer’s comments are listed point by point in Response to the reviewers. Please see the attachment.

Respond to Referee’s comments

The authors have already corrected the manuscript according to reviewer’s comments as following.

Comments from Reviewer: 2

Point

Comment from referee

Summary of corrections

(yellow highlight)

1

Comment: 1

The authors are encouraged to provide a clear statement in the abstract and last paragraph of the introduction about the novelty of their research. Particularly, it would be important to explicitly state what new contributions are made in this manuscript that have not been published in the existing literature. It is always important to elucidate the precise advancements or novel insights gained from a study. Clarifying this point can bolster the significance of the manuscript and provide greater context for readers to understand the value of this research.

We have rewritten the abstract to highlight our work's key aspects and emphasize our approach's novelty. Specifically, we have placed greater emphasis on using BCZT, a material known for its excellent thermo-optic properties, in developing temperature sensors for electric vehicle battery packs. By doing so, we have made it clear that our study focuses on the application of opto-thermal sensors in the context of electric vehicle battery packs, an emerging research area.

2

Comment: 2

The authors are encouraged to add more comments on the applications of the present paper in the abstract.

The abstract section has been improved and rewritten.

3

Comment: 3

The fact that some plots, such as Figures 9 and 10, consist of four subplots without assigning different letters to represent them makes following the paper a little difficult. The authors are encouraged to address this issue to facilitate explaining the results for the readers.

Figures 9 and 10 have been addressed the issue to facilitate explaining the results for the readers.

4

Comment: 4

Some Figures, such as Figures 9, 10, 16, and 17, include very close curves and need to be modified for readability. For example, using different line types in one picture at the same time as different colors.

Figures 9, 10, 16, and 17 have been modified for readability.

5

Comments: 5

Authors are strongly encouraged to comment on other applications of small-scale structures. References such as below

https://doi.org/10.1007/s42417-022-00828-x

https://doi.org/10.1016/j.rineng.2023.101078

https://doi.org/10.1007/s40435-023-01166-w

We have already added the related works and references as suggested in the introduction part line 52-57.

Reviewer 3 Report

The manuscript is written well. The topic is important and interesting to the community. Hence, I recommend to publish the contribution.

Although I am no native English speaker, the quality of language is good. There is place for some minor corrections, being done while setting of the paper.

Author Response

Dear Editor of Journal of Materials,

Thank you for your letter and the reviewers’ comments concerning our manuscript entitled “Investigating the Effects of Calcine Temperature and Dwell Time on Electroluminescent Films in BCZT Ceramics”. Those comments are very valuable and helpful for us to revise and improve our manuscript. We had studied the comments carefully and made some necessary corrections accordingly. The revised sections are marked in yellow highlight in the revised manuscript. The detailed corrections in the revised manuscript and the response to the reviewer’s comments are listed point by point in Response to the reviewers. Please see the attachment.

Respond to Referee’s comments

The authors have already corrected the manuscript according to reviewer’s comments as following. Please see the attachment.

Comments from Reviewer: 3

Point

Comment from referee

Summary of corrections

(yellow highlight)

1

Comment:

The manuscript is written well. The topic is important and interesting to the community. Hence, I recommend to publish the contribution.

We would like to express our sincere gratitude for your positive feedback and recommendation to publish our manuscript. Your acknowledgement of the quality of our writing and the importance of the topic to the community is truly encouraging.

Reviewer 4 Report

The article is devoted to the study of the effect of calcine temperature and dwell time on electroluminescent films based on BCZT ceramics. The article considered two compositions of ceramics and two types of films based on them. The materials used and the production process of the formulations are described quite clearly. But, as for the experimental part of the work, one can add a description of the installations on which the dependences of the dielectric constant and dielectric losses, as well as XRD patterns, were obtained. On the whole, the article is written quite well, the data are presented clearly, and the conclusions are supported by experimental data. The only thing is that I could not find a reference to Figure 1 in the text.

Author Response

Dear Editor of Journal of Materials,

Thank you for your letter and the reviewers’ comments concerning our manuscript entitled “Investigating the Effects of Calcine Temperature and Dwell Time on Electroluminescent Films in BCZT Ceramics”. Those comments are very valuable and helpful for us to revise and improve our manuscript. We had studied the comments carefully and made some necessary corrections accordingly. The revised sections are marked in yellow highlight in the revised manuscript. The detailed corrections in the revised manuscript and the response to the reviewer’s comments are listed point by point in Response to the reviewers. Please see the attachment.

Respond to Referee’s comments

The authors have already corrected the manuscript according to reviewer’s comments as following.

Comments from Reviewer: 4

Point

Comment from referee

Summary of corrections

(yellow highlight)

1

Comment:

The article is devoted to the study of the effect of calcine temperature and dwell time on electroluminescent films based on BCZT ceramics. The article considered two compositions of ceramics and two types of films based on them. The materials used and the production process of the formulations are described quite clearly. But, as for the experimental part of the work, one can add a description of the installations on which the dependences of the dielectric constant and dielectric losses, as well as XRD patterns, were obtained. On the whole, the article is written quite well, the data are presented clearly, and the conclusions are supported by experimental data. The only thing is that I could not find a reference to Figure 1 in the text.

Thank you for your valuable feedback regarding the article. We appreciate your positive assessment of the clarity of the materials used, the production process of the formulations, and the overall quality of the writing and presentation of the data.

We apologize for the oversight in not referencing Figure 1 in the text. Figure 1 illustrates the composition of the electroluminescent films, showing the different layers involved, including the electrode layer, dielectric layer, phosphor layer, and translucent conductive layer. It provides a visual representation of the structure of the films discussed in the article.

In future revisions of the manuscript, we will ensure to refer to Figure 1 explicitly in the relevant sections of the experimental part to provide better clarity and coherence in the presentation of the results.

Furthermore, we appreciate your suggestion to include a description of the experimental setups used to obtain the dependencies of the dielectric constant, dielectric losses, and XRD patterns. This information would indeed enhance the transparency and reproducibility of the experimental work. In future versions of the article, we will incorporate a detailed description of the experimental setups and procedures used to measure these properties.

Once again, we thank you for your valuable feedback, and we will take your suggestions into account when revising the article to improve its clarity and comprehensiveness.

Round 2

Reviewer 2 Report

accept